# MAGICBRUSH 🖌: A Manually Annotated Dataset for Instruction-Guided Image Editing

**Kai Zhang**[1*]   **Lingbo Mo**[1*]   **Wenhu Chen**[2]   **Huan Sun**[1]   **Yu Su**[1]
[1]The Ohio State University   [2]University of Waterloo
{zhang.13253, mo.169, su.809}@osu.edu
https://osu-nlp-group.github.io/MagicBrush

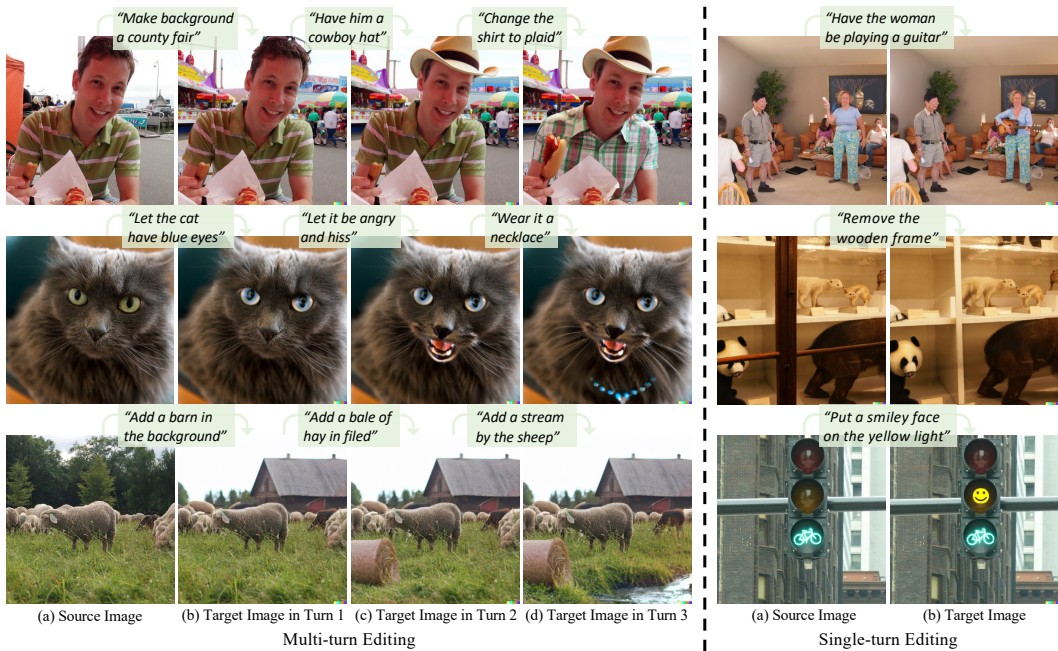

Figure 1: MAGICBRUSH provides 10K manually annotated real image editing triplets (source image, instruction, target image), supporting both single-turn and multi-turn instruction-guided editing.

## Abstract

Text-guided image editing is widely needed in daily life, ranging from personal use to professional applications such as Photoshop. However, existing methods are either zero-shot or trained on an automatically synthesized dataset, which contains a high volume of noise. Thus, they still require lots of manual tuning to produce desirable outcomes in practice. To address this issue, we introduce MAGICBRUSH (https://osu-nlp-group.github.io/MagicBrush/), the first large-scale, manually annotated dataset for instruction-guided real image editing that covers diverse scenarios: single-turn, multi-turn, mask-provided, and mask-free editing. MAGICBRUSH comprises over 10K manually annotated triplets (source image, instruction, target image), which supports trainning large-scale text-guided image editing models. We fine-tune InstructPix2Pix on MAGICBRUSH

---

*Equal Contribution.

and show that the new model can produce much better images according to human evaluation. We further conduct extensive experiments to evaluate current image editing baselines from multiple dimensions including quantitative, qualitative, and human evaluations. The results reveal the challenging nature of our dataset and the gap between current baselines and real-world editing needs.

# 1 Introduction

Applying non-trivial semantic edits to real photos has long been an interesting task in image processing [27]. With the ever-increasing demand for visual content, image editing has become even more essential for enhancing and manipulating images in various fields including photography, advertising, and social media. Natural language, as our innate and flexible interface, serves as an easy way to guide the image editing process. As a result, text-guided image editing [21, 3, 8, 16, 14] has recently gained more popularity compared to other mask-based image editing techniques [20, 35, 23].

Many text-guided image editing methods have been proposed recently and achieved impressive results. These methods can be roughly divided into two categories: (1) zero-shot editing [2, 1, 24], these pipeline methods require massive amount of manual tuning of its hyperparameters to produce reasonable results. (2) end-to-end editing trained on synthetic datasets [4, 37, 7]. However, such silver training data may not only contain annotation errors but also not well capture the need and diversity of real-world editing cases, leading to models with limited editing and generalization abilities.

Therefore, there is an urgent need for a high-quality dataset to facilitate real-world text-guided image editing. In this paper, we present MAGICBRUSH, a large-scale and manually annotated dataset for instruction-guided real image editing. We adopt natural language instruction [29, 4, 41, 22] for its flexibility, which enables users to easily express desired edits with phrases like *"Remove the crowd in the background"* or others shown in Figure 1. Additionally, we extend the dataset to include the multi-turn scenario considering the editing could be conducted iteratively on an image in practice.

We employ a rigorous training and selection for crowd workers, where they need to pass a qualification quiz and undergo manual grading during a trial period. Ongoing spot-checks ensure consistent quality, and failure to maintain high standards results in elimination from the task as shown in Figure 2. During the task, qualified workers need to propose edit instructions and utilize the DALL-E 2 [31] image editing platform to interactively synthesize target image. They will interact with the DALL-E 2 platform with different prompts and hyperparameters until they harvest their desired outputs, otherwise, the example will be dropped. Workers may perform continuous edits on the input image, leading to a series of edit turns. Each turn has a source image (may be the original or output from the previous turn), an instruction, and a target image. We refer to such a complete edit process on a real input image as an edit session. Eventually, we manually check the generated images to ensure quality. MAGICBRUSH consists of 5,313 sessions and 10,388 turns, supporting various editing scenarios including single-/multi-turn, mask-provided, and mask-free for both training and evaluation.

Experiments show that an end-to-end editing method InstructPix2Pix [4], delivers much better results after fine-tuning on MAGICBRUSH and outperforms other baselines according to human preferences. Furthermore, we conduct extensive experiments to evaluate current editing methods from multiple dimensions including quantitative, qualitative, and human evaluations. All these results reveal the challenging nature of MAGICBRUSH and the gap between existing methods and real-world editing needs, calling for more advanced model development in the future.

# 2 Related Work

## 2.1 Text-guided Image Editing

Editing real images has long been an essential task in the field of image processing [27] and recent text-guided image editing has drawn considerable attention. Specifically, it can be categorized into three types in terms of different forms of text.

**Global Description-guided Editing.** Previous methods build fine-grained word and image region alignment for image editing [9, 17, 18]. Recently, Prompt2Prompt [14] modifies words in the original prompts to perform both local editing and global editing by cross-attention control. With the re-

Table 1: Comparison of different image editing datasets. Flower and Bird are domain-specific datasets with global descriptions of target images. EditBench adopts masks (white regions) and local descriptions as guidance, and the size (240) may be insufficient for training. Due to the automatic synthesis process, InstructPix2Pix may contain failure cases.

| Datasets | Real Image? | Open-domain? | Multi-turn? | # Edits | Source | Example Text | Target |
|---|---|---|---|---|---|---|---|
| Oxford-Flower [26] | ✓ | ✗ | ✗ | 8,189 | | *"numerous pale yellow petals and green pedicel with green oval leaves"* | |
| CUB-Bird [38] | ✓ | ✗ | ✗ | 11,788 | | *"this is a grey bird with a brown and yellow tail wing and a red head"* | |
| EditBench [37] | ✓ | ✓ | ✗ | 240 | | *"a flat, dark-colored skateboard with yellow wheels"* | |
| InstructPix2Pix [4] | ✗ | ✓ | ✗ | 313,010 | | *"add a cat"* | |
| MAGICBRUSH | ✓ | ✓ | ✓ | 10,388 | | *"make the man ride a motorcycle"* | |

weighting technique, follow-up work Null Text Inversion [24] further removes the need of original caption for editing by optimizing the inverted diffusion trajectory of the input image. Imagic [16] optimizes a text embedding that aligns with the input image, then interpolates it with the target description, thus generating correspondingly different images for editing. In addition, Text2LIVE [2] trains a model to add an edit layer and combines the edit layer and input image to enable local editing. For global description-guided editing, generally CLIP [30] can be applied to rank generated images *w.r.t* the alignment, thereby delivering higher-ranked results. However, the requirement for detailed descriptions of the target image poses an inconvenience for users.

**Local Description-guided Editing.** Another line of work utilizes masked regions and corresponding regional descriptions for local editing. Blended Diffusion [1] blends edited areas with the other parts of the image at different noise levels along the diffusion process. Imagen Editor [37] trains diffusion editing models by inpainting the masked objects. Local description-guided editing enables fine-grained control by using masks and preserves the other areas intact. However, this method places a greater burden on users, as they must provide additional masks. Also, this approach may be complicated for certain editing types, such as object removal due to the difficulty of describing missing elements.

**Instruction-guided Editing.** Another form of text is instruction, which describes which aspect and how an image should be edited, such as *"change the season to spring"*. Instruction-guided editing, as initially proposed in various studies [11, 13, 42], enables users to edit images without requiring elaborate descriptions or region masking. With advancements in instruction following [29] and image synthesis [15], InstructPix2Pix [4] and SuTI [7] learn to edit images using instructions. Trained with synthetic texts by fine-tuned GPT-3 and images by Prompt2Prompt [14], InstructPix2Pix enables image editing by following instructions. Later work HIVE [41] introduces more training triplets and human ranking results to provide stronger supervision signals for better model training.

## 2.2 Image Editing Datasets

Table 1 compares various semantic editing datasets. Prior work [9, 40, 39, 17, 18] repurposes close-domain image caption datasets [26, 38, 32] for image editing. However, these datasets primarily focus on specific categories like birds and flowers, resulting in limited generalization abilities for the models trained on them. In contrast, open-domain editing meets real-world needs, but high-quality data for training are scarce and challenging to obtain. Although large-scale silver data can be automatically synthesized [4], Table 1 shows the quality may not be desired. EditBench [37] is manually curated

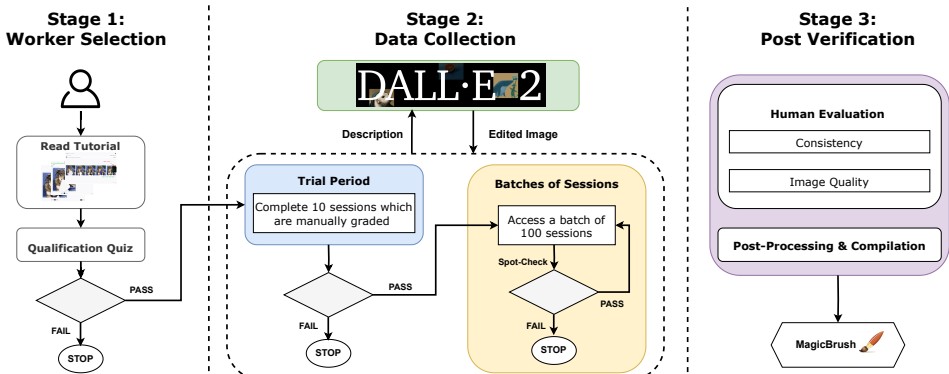

Figure 2: The three-stage crowdsourcing workflow designed for dataset construction.

while it includes only 240 examples, which is insufficient for model training and comprehensive evaluations. Consequently, there is an urgent need for a manually annotated and large-scale dataset.

## 3 MAGICBRUSH Dataset

### 3.1 Problem Definition

Instruction-guided image editing aims to edit a given image following the instruction. In terms of the editing guidance type, this task can be divided into two settings: In **mask-free setting**, given a source image $I_s$ and a textual instruction $T$ of how to edit this image, models are required to generate a target image $I_t$ following the instruction. In **mask-provided setting**, models take an additional free-form mask $M$ to limit the editing region, in addition to the source image and textual instruction. This setting is easier for models but less user-friendly as it requires extra guidance (mask) from users.

Orthogonally, depending on whether the edits are conducted iteratively, we can categorize instruction-guided image editing into two scenarios: single-turn and multi-turn. In **multi-turn scenario**, models take the source image $I_s$ and a sequence of textual instructions $\{T_1, T_2, ..., T_n\}$ to generate intermediate images $\{\widehat{I_{t_1}}, ..., \widehat{I_{t_{n-1}}}\}$ and final image $\widehat{I_{t_n}}$. We term the entire process involving iterative edits as an edit session. The evaluation compares $\widehat{I_{t_n}}$ with the ground truth final image $I_{t_n}$. In **single-turn scenario**, models take both the original source images and intermediate ground truth images $\{I_s, I_{t_1}, ..., I_{t_{n-1}}\}$ as input, editing them only once with corresponding instructions to have $\{\widetilde{I_{t_1}}, \widetilde{I_{t_2}}, ..., \widetilde{I_{t_n}}\}$, respectively. Note that $\widetilde{I_{t_i}}$ and $\widehat{I_{t_i}}$ are usually different except when $i = 1$ where models take the same source image $I_s$ and instruction $T_1$. For single-turn evaluation, we compare all generated images $\{\widetilde{I_{t_1}}, \widetilde{I_{t_2}}, ..., \widetilde{I_{t_n}}\}$ and ground truths $\{I_{t_1}, I_{t_2}, ..., I_{t_n}\}$ pairwisely.

Among these scenarios, mask-free multi-turn editing is the most user-friendly yet challenging setting. Users can achieve complex editing goals with just textual instructions; however, this requires models to edit images iteratively, which easily leads to error accumulations.

### 3.2 Dataset Annotation Pipeline

We focus on real image editing and sample source images from MS COCO dataset [19] for subsequent annotations. We balance 80 object classes of COCO image to increase diversity, thus reducing the over-representation of the person object while keeping the image diversity. Figure 3a shows the final distribution of MAGICBRUSH, with 34.0% person-included images.

We hire crowd workers on Amazon Mechanical Turk (AMT) to manually annotate images using the DALL-E 2 platform.[2] DALL-E 2 is a highly capable text-guided image synthesis platform that can generate high-quality candidate images for editing purposes. However, it requires expertise in providing specific editing guidance, including both global descriptions and masked regions. To

---

[2]AMT: https://www.mturk.com, DALL-E 2: https://openai.com/product/dall-e-2

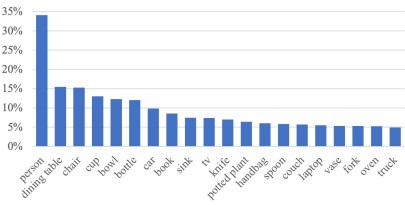

(a) Top 20 object class distribution.

| Number of | Train | Dev | Test | Overall |
|---|---|---|---|---|
| Edit Sessions | 4,512 | 266 | 535 | 5,313 |
| - Sessions with One Edit | 1,789 | 100 | 216 | 2,105 |
| - Sessions with Two Edits | 1,151 | 70 | 120 | 1,341 |
| - Sessions with Three Edits | 1,572 | 96 | 199 | 1,867 |
| Edit Turns | 8,807 | 528 | 1,053 | 10,388 |

(b) Statistics of edit sessions and turns in each data split.

Figure 3: Statistics for the MAGICBRUSH dataset.

ensure the workers could proficiently use the DALL-E 2 platform, we provide them with detailed tutorials, teaching them how to edit images by writing prompts and drawing masks. We employ a stringent worker selection process as shown in Figure 2, and ultimately select 19 workers after thorough filtering. In recognition of the workers' contributions, we spend around $1 for each edit turn, which includes payment for workers on AMT along with the DALL-E 2 platform fees. Qualified workers will interact with DALL-E 2 using various prompts and masks until they achieve desired target images. Please refer to Appendix E for more annotation details.

Specifically, starting from the first edit turn, workers propose a textual instruction $T_1$, its corresponding global description, and a free-form region mask $M_1$ to enable high-quality image synthesis. Then workers try to select the most description-faithful and photo-realistic synthesized image as target image. Note that workers may need to modify their descriptions and masks to find a qualified target image, or even restart with another instruction after several trials. After getting a qualified target image $I_{t_1}$, workers may repeat the annotation process with a new textual instruction $T_2$ based on the current target image $I_{t_1}$ to obtain $I_{t_2}$. In practice, we limit the max number of turns $n$ to 3 for a session, considering workers' possible lack of motivation or inspiration for annotating more turns.

### 3.3 Dataset Analysis and Quality Evaluation

**Data Composition.** Through crowdsourcing, we collect a large-scale instruction-guided image editing dataset named MAGICBRUSH, consisting of over 5K edit sessions and more than 10K edit turns. Figure 3b provides the data splits, as well as the distributions of sessions with varying numbers of edits. Meanwhile, MAGICBRUSH includes a wide range of edit instructions such as object addition/replacement/removal, action changes, color alterations, text or pattern modifications, and object quantity adjustments. The keywords associated with each edit type demonstrate a broad spectrum, covering various objects, actions, and attributes as shown in Figure 4. This diversity indicates that MAGICBRUSH well captures a rich array of editing scenarios, allowing for comprehensive training and evaluation of instruction-guided image editing models.

**Data Quality Evaluation.** We invite five AMT workers to review 500 randomly sampled edit turns from MAGICBRUSH, with each evaluating 100 turns. Given an edit turn (source image, edit instruction, and target image), the worker is required to measure the edited image from two aspects: *consistency* and *image quality*. Consistency evaluates how well the editing to the original image aligns with the instruction. Image quality assesses the overall quality of the edited image, considering factors such as maintaining the visual fidelity of the original image, seamless blending of edited elements with the original image, and the natural appearance of the changes. Workers provide a score between 1 and 5 for each criterion. The average scores for consistency and image quality are reported as 4.1 and 3.9 out of 5.0, respectively. Compared to edited images by existing methods in Section 4.4, these numbers demonstrate the high quality of the MAGICBRUSH dataset.

## 4 Experiments

### 4.1 Experiment Setup

**Baselines.** For comprehensiveness, we consider multiple baselines in both mask-free and mask-provided settings. For all baselines, we adopt the default hyperparameters available in the official

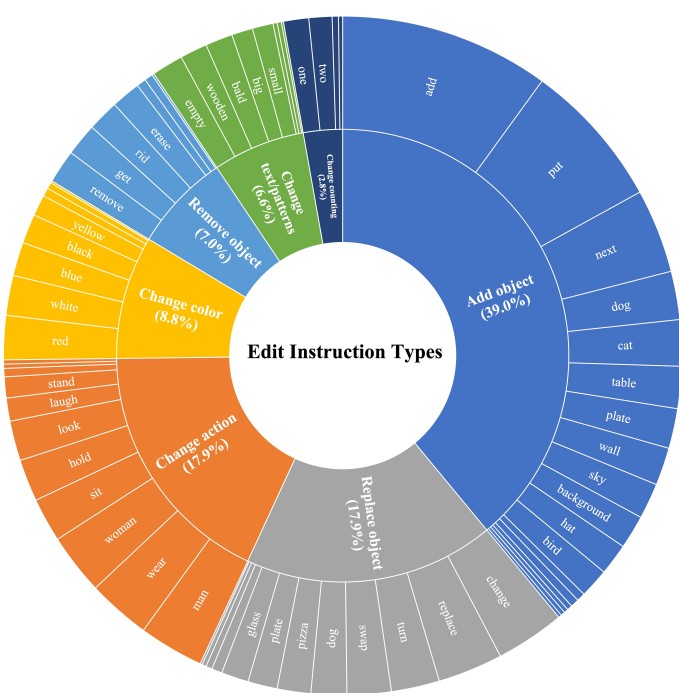

Figure 4: An overview of keywords in edit instructions. The inner circle depicts the types of edits and outer circle showcases the most frequent words used within each type.

code repositories to guarantee reproducibility and fairness. Given that some baselines may require global and local descriptions, inspired by InstructPix2Pix [4], we instruct ChatGPT [28] to generate desired text formats. Please refer to the Appendix C.4 for prompt details. Specifically, for **mask-free editing baselines**, we consider: (1) *Open-Edit* [21], (2) *VQGAN-CLIP* [8], (3) *SD-SDEdit* [23], (4) *Text2LIVE* [2], (5) *Null Text Inversion* [24], (6) *InstructPix2Pix* [4] and its fine-tuned version on the training set of MAGICBRUSH, (7) *HIVE* [41] and its fine-tuned version on MAGICBRUSH. For **mask-provided baselines**, we consider: (1) *GLIDE* [25] and (2) *Blended Diffusion* [1]. Please refer to Appendix C.2 for more implementation and fine-tuning details.

**Evaluation Metrics.** We utilize L1 and L2 to measure the average pixel-level absolute difference between the generated image and ground truth image. In addition, we adopt CLIP-I and DINO, which measure the image quality with the cosine similarity between the generated image and reference ground truth image using their CLIP [30] and DINO [6] embeddings. Finally, CLIP-T [34, 7] is used to measure the text-image alignment with the cosine similarity between local descriptions and generated images CLIP embeddings. We use local description because the global one is not specific to the editing region and the edit instruction may not describe the target image.

## 4.2 Quantitative Evaluation

We evaluate mask-free and mask-provided baselines separately with the same 535 sessions from test set, as the latter requires mask as additional editing guidance, making it relatively easier. For each setting, we consider single- and multi-turn editing scenarios described in Section 3.1.

**Mask-free Editing.** Table 2 shows the results of mask-free methods which are given instructions only to edit images. We have the following observations: (1) In general, all methods perform worse in the multi-turn scenario due to the error accumulation in iterative editing. (2) The off-the-shelf InstructPix2Pix [4] checkpoint is not competitive compared to other baselines, in both single-turn and multi-turn scenarios. However, after fine-tuning on MAGICBRUSH, InstructPix2Pix shows significant performance improvements across all metrics, achieving the best or second-best results under most metrics. Such improvement introduced by MAGICBRUSH is consistent on HIVE [41]. These suggest that instruction-guided image editing models could substantially benefit from training on our MAGICBRUSH dataset, demonstrating its usefulness. (3) Text2LIVE [2] performs well in L1

Table 2: Quantitative study on mask-free baselines on MAGICBRUSH test set. Multi-turn setting evaluates the final target images that iteratively edited on the first source images in edit sessions. The best results are marked in **bold**.

| Settings | Methods | L1↓ | L2↓ | CLIP-I↑ | DINO↑ | CLIP-T↑ |
|---|---|---|---|---|---|---|
| | *Global Description-guided* | | | | | |
| **Single-turn** | Open-Edit [21] | 0.1430 | 0.0431 | 0.8381 | 0.7632 | 0.2610 |
| | VQGAN-CLIP [8] | 0.2200 | 0.0833 | 0.6751 | 0.4946 | **0.3879** |
| | SD-SDEdit [23] | 0.1014 | 0.0278 | 0.8526 | 0.7726 | 0.2777 |
| | Text2LIVE [2] | 0.0636 | **0.0169** | 0.9244 | 0.8807 | 0.2424 |
| | Null Text Inversion [24] | 0.0749 | 0.0197 | 0.8827 | 0.8206 | 0.2737 |
| | *Instruction-guided* | | | | | |
| | HIVE [41] | 0.1092 | 0.0341 | 0.8519 | 0.7500 | 0.2752 |
| | w/ MagicBrush | 0.0658 | 0.0224 | 0.9189 | 0.8655 | 0.2812 |
| | InstructPix2Pix [4] | 0.1122 | 0.0371 | 0.8524 | 0.7428 | 0.2764 |
| | w/ MagicBrush | **0.0625** | 0.0203 | **0.9332** | **0.8987** | 0.2781 |
| | *Global Description-guided* | | | | | |
| **Multi-turn** | Open-Edit [21] | 0.1655 | 0.0550 | 0.8038 | 0.6835 | 0.2527 |
| | VQGAN-CLIP [8] | 0.2471 | 0.1025 | 0.6606 | 0.4592 | **0.3845** |
| | SD-SDEdit [23] | 0.1616 | 0.0602 | 0.7933 | 0.6212 | 0.2694 |
| | Text2LIVE [2] | 0.0989 | **0.0284** | 0.8795 | 0.7926 | 0.2716 |
| | Null Text Inversion [24] | 0.1057 | 0.0335 | 0.8468 | 0.7529 | 0.2710 |
| | *Instruction-guided* | | | | | |
| | HIVE [41] | 0.1521 | 0.0557 | 0.8004 | 0.6463 | 0.2673 |
| | w/ MagicBrush | 0.0966 | 0.0365 | 0.8785 | 0.7891 | 0.2796 |
| | InstructPix2Pix [4] | 0.1584 | 0.0598 | 0.7924 | 0.6177 | 0.2726 |
| | w/ MagicBrush | **0.0964** | 0.0353 | **0.8924** | **0.8273** | 0.2754 |

Table 3: Quantitative study on mask-provided baselines on MAGICBRUSH test set. L1, L2, and CLIP-T are measured over the masked regions only. The best results are marked in **bold**.

| Settings | Methods | L1↓ | L2↓ | CLIP-I↑ | DINO↑ | CLIP-T↑ |
|---|---|---|---|---|---|---|
| **Single-turn** | GLIDE [25] | **3.4973** | **115.8347** | **0.9487** | **0.9206** | 0.2249 |
| | Blended Diffusion [1] | 3.5631 | 119.2813 | 0.9291 | 0.8644 | **0.2622** |
| **Multi-turn** | GLIDE [25] | **11.7487** | **1079.5997** | **0.9094** | **0.8494** | 0.2252 |
| | Blended Diffusion [1] | 14.5439 | 1510.2271 | 0.8782 | 0.7690 | **0.2619** |

and L2 evaluations, likely due to the addition of an extra editing layer that minimizes changes to the source image. As a result, edited images fail to satisfy the instructions, as evidenced by the low CLIP-T score. VQGAN-CLIP [8] achieves the highest CLIP-T score because it fine-tunes the model during inference with CLIP as the direct supervision. However, the edited images may change too significantly, leading to unfavorable results on other metrics.

**Mask-provided Editing.** Table 3 lists the results of two mask-provided methods. As observed in the mask-free setting, the multi-turn scenario is more challenging than the single-turn scenario. While both mask-provided methods achieve high scores under the CLIP-I and DINO metrics, they fail to deliver satisfactory results according to the other three metrics (L1, L2, and CLIP-T) that evaluate local regions. Notably, after tuning on MAGICBRUSH, InstructPix2Pix [4] achieves better editing results than mask-provided Blended Diffusion [1] in terms of CLIP-I and DINO metrics. This suggests that fine-tuning with our data could maintain good image quality.

## 4.3 Qualitative Evaluation

We present the results of the top-performing mask-free (Text2LIVE [2]) and mask-provided (GLIDE [25]) methods in our qualitative analysis. We also compare the original and fine-tuned checkpoints of InstructPix2Pix [4]. Figure 5 illustrates the iterative results of these four models

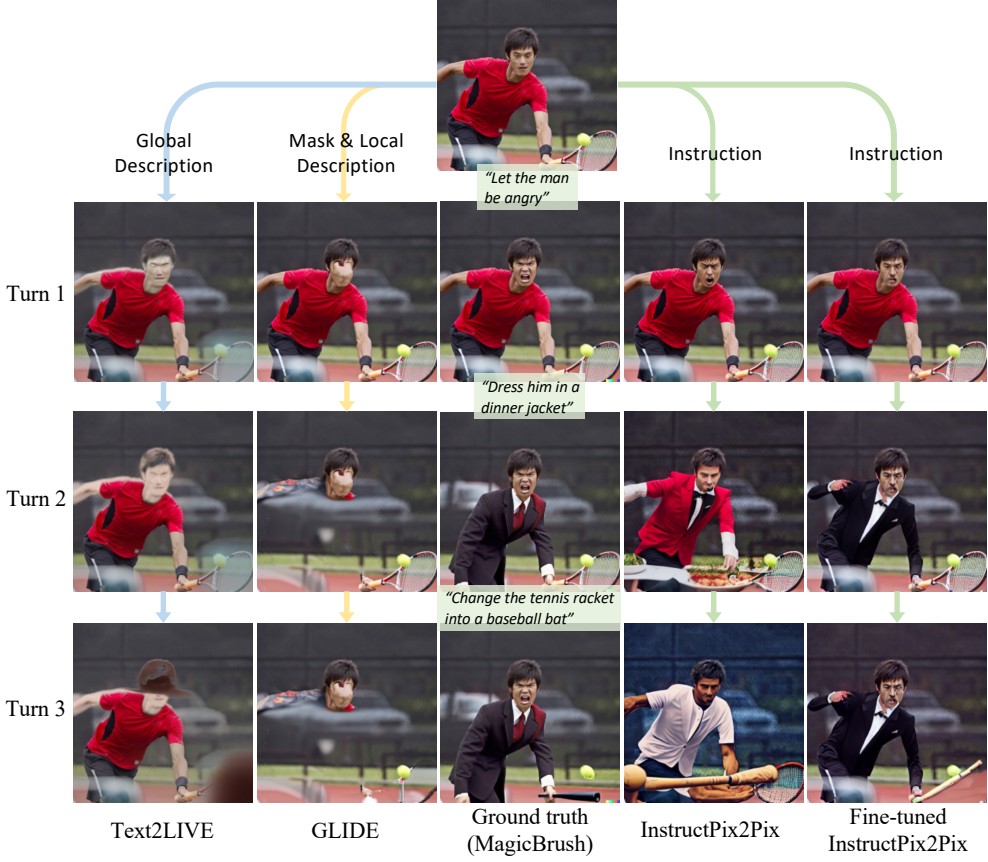

Figure 5: Qualitative evaluation of multi-turn editing scenario. We provide all baselines their desired input formats (e.g., masks and local descriptions for GLIDE).

and ground truth images from MAGICBRUSH. Both Text2LIVE and GLIDE are unsuccessful in editing the man's face and clothes. The original InstructPix2Pix changes the images following the instructions; however, the resulting images exhibit excessive modification and lack photorealism. Fine-tuning InstructPix2Pix on MAGICBRUSH alleviates this issue, but the images remain notably inferior to the ground truth ones. Please see Appendix D for more examples of qualitative evaluation.

## 4.4 Human Evaluation

We conduct comprehensive human evaluations to assess both *consistency* and *image quality* on generated images. Our evaluations encompass three tasks: multi-choice image comparison, one-on-one comparison, and individual image evaluation. We randomly sample 100 image examples from test set for each task and hire 5 AMT workers as evaluators to perform the tasks. For each task, the images are evenly assigned to evaluators and the averaged scores (if applicable) are reported.

**Multi-choice Comparison.** The multi-choice comparison involves four top-performing methods in Table 2 and Table 3, including Text2LIVE, GLIDE, InstructPix2Pix, and fine-tuned InstructPix2Pix on MAGICBRUSH. For each example, evaluators need to select the best edited image based on consistency and image quality, respectively. The results in Table 4 indicate that fine-tuned InstructPix2Pix attains the highest performance, significantly surpassing the other three methods. This outcome validates the effectiveness of training on our MAGICBRUSH dataset. Interestingly, while Text2LIVE achieves a high score in auto evaluation, its performance in human evaluation appears to be less desirable, especially in terms of the instruction consistency. This indicates current automatic metrics that focus on the overall image quality may not align well with human preferences, emphasizing the need for future research to develop better automatic metrics.

Table 4: Multi-choice comparison of four methods. The numbers represent the frequency of each method being chosen as the best for each aspect.

| | Text2LIVE [2] | GLIDE [25] | InstructPix2Pix [4] | Fine-tuned InstructPix2Pix |
|---|---|---|---|---|
| Consistency | 0 | 16 | 33 | **51** |
| Image Quality | 9 | 15 | 27 | **49** |

Table 5: One-on-one comparisons between fine-tuned InstructPix2Pix and other methods including InstructPix2Pix and Text2LIVE, as well as ground truth (GT). The numbers in the table indicate the frequency of each method being chosen as the better option. To account for scenarios where two methods perform equally, we include a "Tie" option in each question for comprehensive evaluation.

| Settings | Consistency | | | Image Quality | | |
|---|---|---|---|---|---|---|
| **Single-turn** | Fine-tuned InstructPix2Pix **40** | InstructPix2Pix [4] 35 | Tie 25 | Fine-tuned InstructPix2Pix **48** | InstructPix2Pix [4] 33 | Tie 19 |
| | Fine-tuned InstructPix2Pix **68** | Text2LIVE [2] 4 | Tie 28 | Fine-tuned InstructPix2Pix **61** | Text2LIVE [2] 19 | Tie 20 |
| **Multi-turn** | Fine-tuned InstructPix2Pix 13 | GT (Turn 1) **72** | Tie 15 | Fine-tuned InstructPix2Pix 19 | GT (Turn 1) **64** | Tie 17 |
| | Fine-tuned InstructPix2Pix 13 | GT (Turn 2) **80** | Tie 7 | Fine-tuned InstructPix2Pix 19 | GT (Turn 2) **60** | Tie 21 |
| | Fine-tuned InstructPix2Pix 11 | GT (Turn 3) **80** | Tie 9 | Fine-tuned InstructPix2Pix 6 | GT (Turn 3) **75** | Tie 19 |

**One-on-one Comparison.** The one-on-one comparison provides a detailed and nuanced evaluation of the fine-tuned InstructPix2Pix by comparing it against strong baselines and ground truth. Evaluators are asked to determine the preferred option based on consistency and image quality, respectively. We divide the comparisons into two scenarios as mentioned in Section 3.1: (1) In the single-turn scenario, we compare fine-tuned InstructPix2Pix and two other methods (InstructPix2Pix and Text2LIVE). As shown in Table 5, fine-tuned InstructPix2Pix consistently outperforms the other two methods in terms of both consistency and image quality. (2) In the multi-turn scenario, we compare the fine-tuned InstructPix2Pix with ground truth images to observe how the quality of edited images varies across different turns. The results reveal that the performance gap generally widens as the number of edit turn increases. This finding highlights the challenges associated with error accumulation in current top-performing models and underscores the difficulties posed by our dataset.

**Individual Evaluation.** The individual evaluation employs a 5-point Likert scale to measure the quality of individual images generated by four specific models, gathering subjective user feedback. Evaluators are asked to rate the images on a scale from 1 to 5, assessing both consistency and image quality. Each evaluator receives an equal share of the images, specifically evaluating 80 images in total, with 20 images from each of the four models. The results in Table 6 clearly demonstrate that fine-tuned InstructPix2Pix outperforms Text2LIVE and GLIDE, and further improves upon the performance of InstructPix2Pix. This finding highlights the advantages of training or fine-tuning models using the MAGICBRUSH dataset.

Table 6: Individual evaluation using a 5-point Likert scale. The numbers in the table represent the average scores calculated for each aspect.

| | Consistency | Image Quality |
|---|---|---|
| Text2LIVE [2] | 1.1 | 2.8 |
| GLIDE [25] | 1.8 | 2.8 |
| InstructPix2Pix [4] | 3.0 | 3.2 |
| Fine-tuned InstructPix2Pix | **3.1** | **3.6** |

## 5 Conclusion and Future Work

In this work, we present MAGICBRUSH, a large-scale and manually annotated dataset for instruction-guided real image editing. Although extensive experiments show that InstructPix2Pix fine-tuned on

MAGICBRUSH achieves the best results, its edited images are still notably inferior compared to the ground truth ones. This observation indicates the effectiveness of our dataset for training and the gap between current methods and real-world editing needs. We hope MAGICBRUSH will contribute to the development of more advanced models and human-preference-aligned evaluation metrics for instruction-guided real image editing in the future.

## Acknowledgements

The authors would like to thank colleagues from the OSU NLP group for their constructive feedback, Yuxuan Sun for discussing the fine-tuning of InstructPix2Pix, and the contributors from the Amazon Mechanical Turk platform for their participation in the study and assistance with data collection. This research was sponsored in part by NSF CAREER #1942980, ARL W911NF2220144, NSF OAC 2112606, and NSF OAC 2118240. The views and conclusions contained herein are those of the authors and should not be interpreted as representing the official policies, either expressed or implied, of the U.S. government. The U.S. Government is authorized to reproduce and distribute reprints for Government purposes notwithstanding any copyright notice herein.

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

# Appendices

## A  Overview

Our supplementary includes the following sections:

- **Section B: Discussions.** Discussions of Limitations, Alleviating Potential Model Bias, Social Impacts, Ethical Considerations, and License of Assets.
- **Section C: Implementation Details.** Details for implementing baselines and fine-tuning InstructPix2Pix with MAGICBRUSH.
- **Section D: More Qualitative Study.** More qualitative study including both single-turn and multi-turn scenarios.
- **Section E: Data Annotation.** Details for dataset collection and image quality evaluation.

We share the following artifacts:

Table 7: Shared artifacts in this work, we protect the test split with a password to avoid web crawling for model training.

| Artifact | Link | License |
|---|---|---|
| Homepage | `https://osu-nlp-group.github.io/MagicBrush/` | - |
| Code Repository | `https://github.com/OSU-NLP-Group/MagicBrush` | CC BY 4.0 |
| Training and Dev Set | `https://huggingface.co/datasets/osunlp/MagicBrush` | CC BY 4.0 |
| Test Set | `https://shorturl.at/alHMO` (Password: MagicBrush) | CC BY 4.0 |

# B    Discussions

## B.1    Limitations

Although our image annotation is based on a lot of manual effort and conducted on the powerful editing platform (DALL-E 2), a small portion of edits ($<5\%$) may contain minor extra modifications that are not mentioned by the instruction or may still look slightly unnatural to some individuals. That being said, we believe it would not affect the overall quality of MAGICBRUSH as the experiments have shown that our dataset can largely enhance the model's abilities of editing real images *w.r.t* the given instruction.

While MAGICBRUSH supports various edit types on real images, it does not contain data for global editing (e.g., style transfer) due to annotation built upon DALL-E 2. However, such edit turn could be easily obtained automatically [4] due to its less photorealism and more artistic nature.

## B.2    Alleviating the Potential Model Bias

After conducting an in-depth pilot exploration on various generative models, including commercial image-editing platforms, we have found that DALL-E 2 is one of the best available editing models. It is highly likely that users can obtain satisfactory images that meet their editing goals, provided they carry out sufficient trials on the prompting and masking. However, solely using one model for ground truth generation may result in the potential bias inherent in that model.

To alleviate this, we adopt the following strategies from two aspects: 1) Diversity of instruction: Through clear guidance in our tutorial and frequent communication via email, we strongly encourage workers to design diverse instruction. In practice, we reject some repetitive or trivial edits and suggest alternatives to ensure the diversity. 2) Diversity of images: We carefully design a sampling strategy to ensure the objects in the images are more balanced and decrease the chance of sampling simple images with fewer objects. In this way, the editing largely varies since the edited regions are required to be naturally blended with the context. With these efforts, MAGICBRUSH has less recurring edit patterns and higher diversity, thus minimizing potential biases.

That being said, admittedly, it is challenging to eliminate the inherent biases completely. We commit to remaining alert for potential biases in our dataset identified by the community, and will take prompt rectification actions.

## B.3    Social Impacts

MAGICBRUSH has the potential to significantly improve the capabilities of text-guided image editing systems, enabling a broader range of users to easily manipulate images. On one hand, this could lead to numerous positive social impacts: users can achieve their editing goals through instructions alone, without the need for professional editing knowledge, such as using Photoshop or painting. Such an effortless editing process can save users' time spent on manual operation, resulting in increased efficiency. Furthermore, it can facilitate image creation and manipulation for users with visual or motor impairments, given they can rely on language instructions as input.

On the other hand, the potential risks associated with such advanced image editing systems deserve attention. Malicious users could exploit editing tools to create realistic fake or harmful content, leading to the spread of misinformation. It is essential to implement appropriate safeguards and responsible AI frameworks when developing user-friendly image editing systems.

## B.4    Ethical Considerations

The COCO [19] dataset focuses on common objects and context, rather than specific people or places. In our annotation guidelines, we also forbid annotators from creating any identifiable information (e.g., human faces). Furthermore, DALL-E 2 adheres to strict rules to exclude prompts related to harmful, inappropriate, or sensitive content. As a result, MAGICBRUSH inherently minimizes the potential for privacy or harmful concerns as it relies on images sourced from the COCO dataset and annotations built upon DALL-E 2.

To ensure the collection of high-quality data and fair treatment of our crowdworkers, we have implemented a meticulous payment plan for the AMT task. We conduct a pilot study to estimate the

average time required to complete a session. It reveals that the duration ranges from 4 to 8 minutes, depending on the number of edit turns performed by the workers within each session. This results in a total annotation time of approximately 529 worker hours. This information also allows us to appropriately adjust the payment, ensuring it exceeds the minimum wage amount in our state. As a result, we offer an initial payment of 80 cents for the first edit turn in each session, along with a bonus of 40 cents for each additional edit turn within the same session. This allows workers to potentially earn up to $1.6 per session, encouraging their active participation and rewarding their efforts accordingly. In total, the cost of creating the MAGICBRUSH dataset amounts to approximately $11,000 which includes the payments made on AMT ($8,000) and DALL-E 2 API ($3,000) costs.

### B.5 License of Assets

For baselines, VQGAN-CLIP [8], Text2LIVE [2], and Blended Diffusion [1] are under the MIT License. SD-SDEdit [33, 23] is released under the Creative ML OpenRAIL-M License, and Instruct-Pix2Pix [4] inherits this license as it is built upon Stable Diffusion. Null Text Inversion [24] and GLIDE [25] are under the Apache-2.0 License.

For dataset, COCO [19] is under Creative Commons Attribution 4.0 License. According to DALL-E 2, we own the images created with DALL-E 2, including the right to reprint, sell, and merchandise. We decide to release MAGICBRUSH under Creative Commons Attribution 4.0 License for easy access in the research community. The license allows users to share and adapt the dataset for any purpose, even commercially, as long as appropriate credit is given and any changes made are indicated. By providing the dataset under this license, we hope to encourage researchers and practitioners to explore and advance the field of text-guided image editing further.

## C  Implementation Details

### C.1  COCO Image Sampling

Given the highly unbalanced distribution of objects in COCO, where 54.2% of images contain a person, we employ a class-balanced sampling strategy for the 80 classes. In particular, for each class, we select one image containing an object from the target class, ensuring it has no overlap with the current image pool. This process is repeated as we move through each class. Notably, one COCO image may contain multiple objects from different classes, so it is possible to sample images with a person for non-person classes. To mitigate the over-representation of person class, we prioritize selecting images without a person for non-person classes by reducing the sampling probability of images containing a person by half.

### C.2  Baseline Details

For all baselines, we adopt the default hyperparameters available in the official code repositories to guarantee reproducibility and fairness. Specifically, for **mask-free editing baselines**, we consider:

(1) *Open-Edit* [21] is a GAN-based method pre-trained with reconstruction loss and fine-tuned on the given image with consistency loss. It edits image by performing arithmetic operations on word embeddings within a shared vector space with visual features.

(2) *VQGAN-CLIP* [8] fine-tunes VQGAN [12] with CLIP embedding [30] similarity between generated image and target text. Then it generates the image with the optimized VQGAN embedding.

(3) *SD-SDEdit* [23] is a tuning-free method built upon Stable Diffusion [33]. Based on the target description, SDEdit adds stochastic differential equation noise to the source image and then denoises the target image through that prior.

(4) *Text2LIVE* [2] fine-tunes Vision Transformer [10] to generate the edited object on the extra edited layer with data augmentation and CLIP [30] supervision. The target image is the composite of the extra edit layer and the original layer.

(5) *Null Text Inversion* [24] optimizes DDIM [36] trajectory to restore the source image and then performs image editing on the denoising process with text-image cross-attention control [14].

(6) *InstructPix2Pix* [4] is pre-trained with automatically curated instruction-following editing data, initialized from Stable Diffusion [33]. It edits the source image by controlling the faithfulness to instruction and similarity with the source image, without any test-time tuning.

(7) *HIVE* [41] is trained with more data synthesized using a method similar to InstructPix2Pix [4] and is further fine-tuned with a reward model trained with human-ranked data.

For **mask-provided baselines**, we consider:

(1) *GLIDE* [25] is trained with 67M text-image pairs where all images are person-free. To edit, it fills in the masked region of an image conditioned on the local description with CLIP [30] guidance.

(2) *Blended Diffusion* [1] resorts to CLIP [30] guidance during a masked region denoising process and blends it with the context in the noisy source image at each denoising timestep to increase the region-context consistency of the generated target image.

## C.3  InstructPix2Pix Fine-tuning Details.

We continually fine-tune the checkpoint with the training set of MAGICBRUSH. Specifically, we train 168 epochs on $2 \times$ 40GB NVIDIA A100 GPUs with a total batch size of 64. Following prior work [4], we use a $256 \times 256$ image resolution and the same training strategies and hyper-parameters.

## C.4  ChatGPT Prompts

Table 8: Prompts on ChatGPT for global and local description generation.

| | |
|---|---|
| **Global Description** | Given the original caption and a edit instruction, write a caption after editing.

Original Caption: Painting of The Flying Scotsman train at York station
Edit Instruction: add airplane wings
Final Caption: Painting of The Flying Scotsman train with airplane wings at York station

Original Caption: Old Boat at Sunderland Point by Steve Liptrot
Edit Instruction: remove the boat
Final Caption: Empty Sunderland Point by Steve Liptrot

Original Caption: "Charles Lindbergh ""Spirit of St. Louis"""
Edit Instruction: have it be about Beijing
Final Caption: "Charles Lindbergh ""Spirit of Beijing"""

Original Caption: [CAPTION]
Edit Instruction: [INSTRUCTION]
Final Caption: |
| **Local Description** | Given the original caption and an edit instruction, write a local short description for specific location to describe the object. If it's removing, leave it blank.

Original Caption: Painting of The Flying Scotsman train at York station
Edit Instruction: add airplane wings
Local Caption: airplane wings

Original Caption: Old Boat at Sunderland Point by Steve Liptrot
Edit Instruction: remove the boat
Local Caption:

Original Caption: A demonic looking chucky like doll standing next to a white clock.
Edit Instruction: Make the doll wear a hat
Local Caption: hat

Original Caption: [CAPTION]
Edit Instruction: [INSTRUCTION]
Local Caption: |

To transform the edit instruction to global description and local description required by other baselines and facilitate future research. Inspired by InstructPix2Pix [4], we instruct ChatGPT (davinci-turbo-0301) to generate the target text formats given the input image caption and instruction. Specifically, as shown in Tab 8, we provide clear instructions and three in-context learning examples [5] for ChatGPT to learn the generation rules, thus generating the desired text formats for baselines.

## D  More Qualitative Study

Figure 7 shows the results of top-performing baselines in multi-turn editing scenarios. And the observation is consistent with that shown in Figure 5.

In addition, we show more baselines in the single-turn editing scenario in Figure 6. Even in such a relatively easier scenario, most baselines fail to edit precisely. Although InstructPix2Pix edits the images following the instruction to some extent, it tends to modify the images too much, resulting in the loss of some important details or incorrect changes.

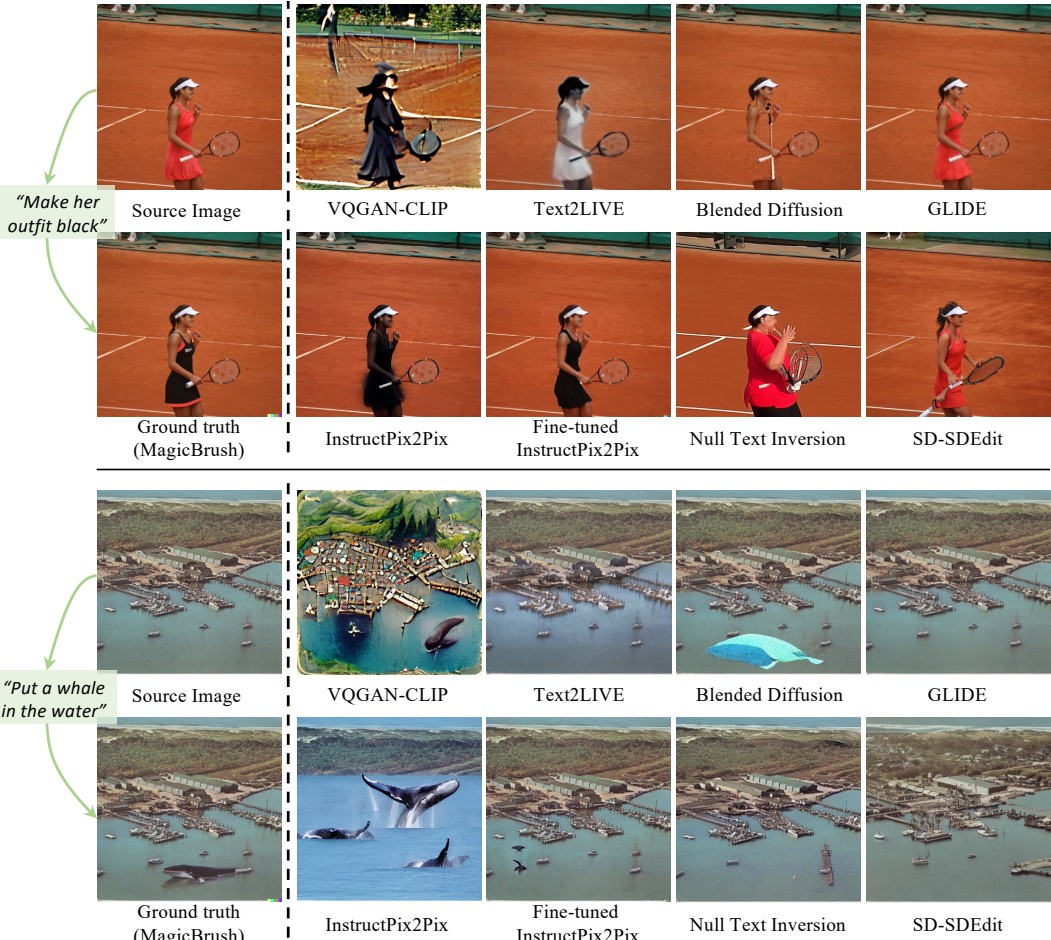

Figure 6: Qualitative evaluation of single-turn editing scenario. We provide all baselines their desired input formats (e.g., masks and local descriptions for Blended Diffusion and GLIDE).

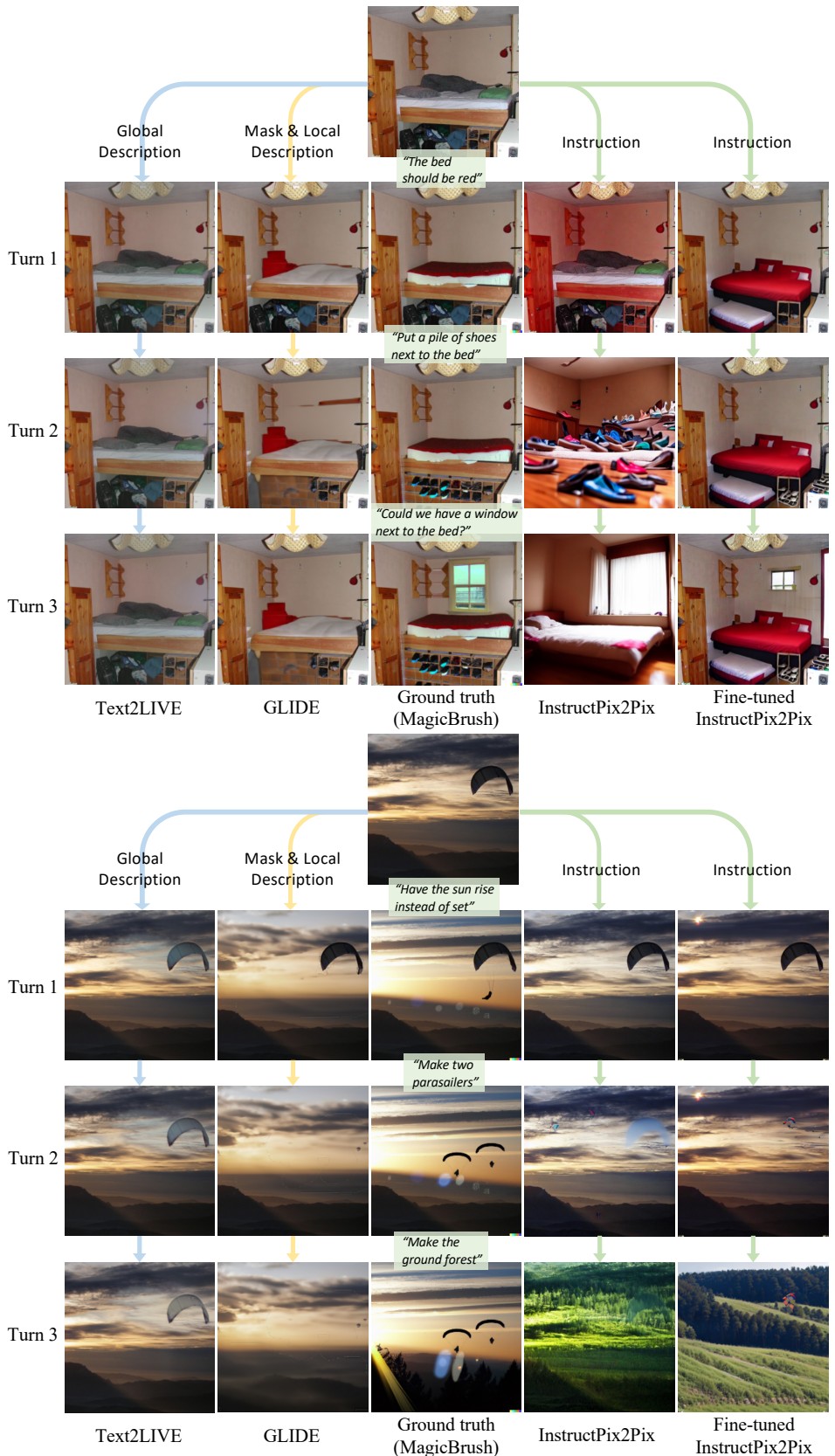

Figure 7: Qualitative evaluation of multi-turn editing scenario. We provide all baselines their desired input formats (e.g., masks and local descriptions for GLIDE).

# E  Data Annotation

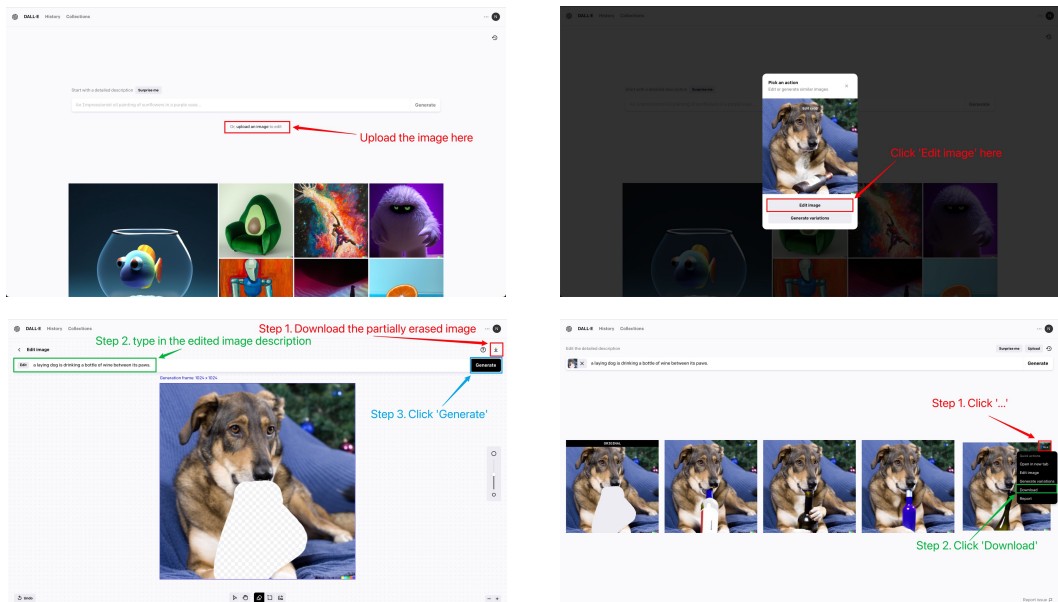

Figure 8: Illustration of the step-by-step instructions in annotation tutorial.

## E.1  Annotation Tutorial

We conduct the data collection and deploy the interfaces on AMT. Our approach entails a meticulous design of the entire process to streamline the procedure and enhance its efficiency. To facilitate workers understanding and proper execution of the data annotation, we provide them with an elaborate tutorial contained in a 5-page document (`https://shorturl.at/bpBUW`), along with a supplementary video demonstration (`https://www.youtube.com/watch?v=husejlhNyfo`). These links remain accessible at all times for reference purposes.

In the tutorial, we ensure that each step of the interface is accompanied by detailed instructions, making it self-contained and easy to follow. Figure 9 displays the interfaces used in our crowdsourcing task for data collection, offering a visual representation of the user experience.

The annotation is divided into three phases: *Preparation*, *Initial Editing*, and *Follow-up Editing*. In the Preparation phase, we provide clear instructions on how to access the source image, log in to DALL-E 2, and upload the source image to prepare for editing.

During the Initial Editing phase, we clarify the terms "Edit Instruction" and "Global Description", ensuring workers understand their respective purposes.

- "Edit Instruction" is a directive that describes the suggested edits and how workers wish to alter the image. We encourage workers to phrase their instructions as if they are speaking to a helper in a simple and colloquial manner, such as 'Let the dog drink the wine'.

- "Global Description" provides a comprehensive description of the image after the suggested edit has been applied, e.g., 'A dog lying down is holding a bottle of wine between its paws'. This description is input into DALL-E 2 to generate the desired image. We also specify the expected outcomes of this initial edit to guarantee all steps are covered and prevent any omissions.

In the Follow-up Editing phase, users are free to carry out follow-up edit turns on the image generated in the first turn. The process remains similar to the second phase, facilitating a smooth continuation of the annotation process.

### E.2 Monitoring the Annotation Process

Throughout the task, workers are encouraged to provide comments and feedback after each session. Also, during the entire annotation process, we continuously check the data to ensure the quality. Specifically, in the trial period, we checked all annotated examples in a batch with 10 sessions to provide prompt feedback to each worker on data quality (both in image and instruction). Only workers that can deliver satisfactory results will be advanced to the next stage, where they will be asked to do more tasks on AMT. Then, we spot checked on 5 of each 100 sessions in the rest of the annotation process. In checking, the sessions containing subpar images, with issues relating to image quality and instruction consistency, are eliminated. Additionally, we maintained frequent communication with the workers, providing timely guidance and requesting certain turns to be redone if the quality is unsatisfactory. As time progresses, we observe a significant decrease in the frequency of communication, and we find that all workers consistently pass the checks in the later batches of data annotations. This indicates a notable improvement in the quality of the annotated data as the process advances.

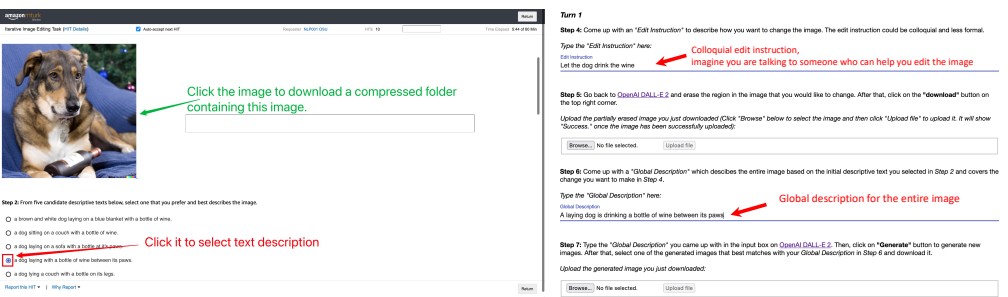

Figure 9: Data collection interface on AMT.

### E.3 Human Evaluation

We conduct multiple human evaluation tasks on AMT to assess the quality of our dataset (Section 3.3) and evaluate the generated images from different models (Section 4.4). For these tasks, we design three different types of interfaces. The first type (Figure 10) involves individual evaluation using a 5-point Likert scale to measure the quality of the images. The second type (Figure 11) is a multi-choice comparison task, where evaluators compare four top-performing methods in Table 2 and Table 3, including Text2LIVE, GLIDE, InstructPix2Pix, and fine-tuned InstructPix2Pix on MAGICBRUSH. The last type (Figure 12) is a one-on-one comparison task, providing a more nuanced evaluation between fine-tuned InstructPix2Pix and other strong baselines as well as the ground truth. Both consistency and image quality are assessed in each human evaluation task, with the original image and the textual instruction provided at the beginning.

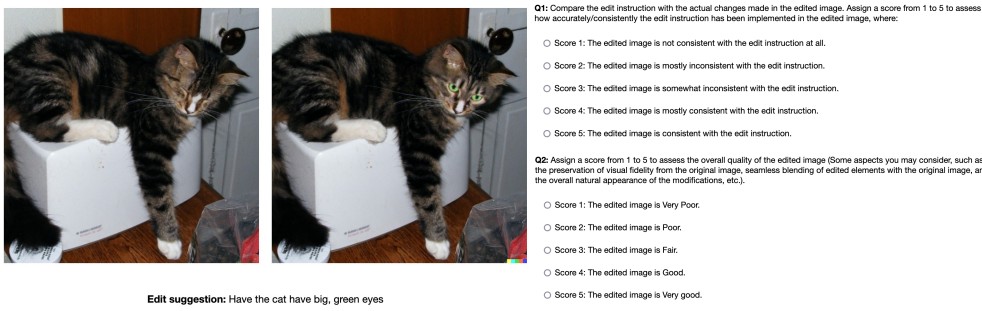

Figure 10: The interface of individual evaluation on AMT to assess the dataset quality as well as generated images by different models.

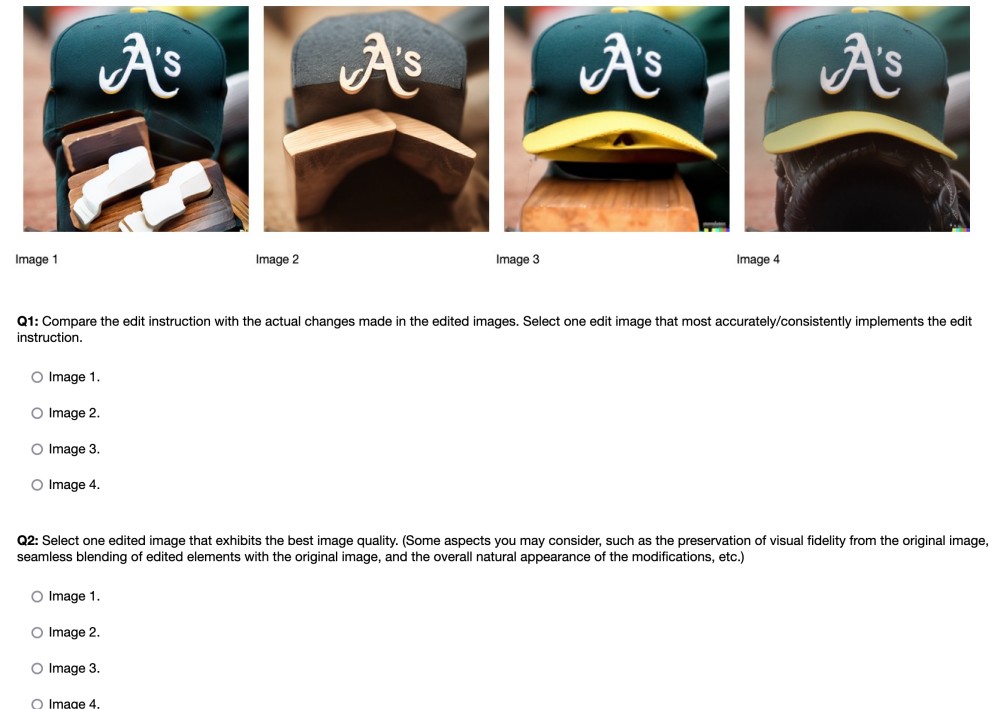

**Q1:** Compare the edit instruction with the actual changes made in the edited images. Select one edit image that most accurately/consistently implements the edit instruction.

○ Image 1.

○ Image 2.

○ Image 3.

○ Image 4.

**Q2:** Select one edited image that exhibits the best image quality. (Some aspects you may consider, such as the preservation of visual fidelity from the original image, seamless blending of edited elements with the original image, and the overall natural appearance of the modifications, etc.)

○ Image 1.

○ Image 2.

○ Image 3.

○ Image 4.

Figure 11: The interface of multi-choice comparison on AMT to evaluate generated images by different models.

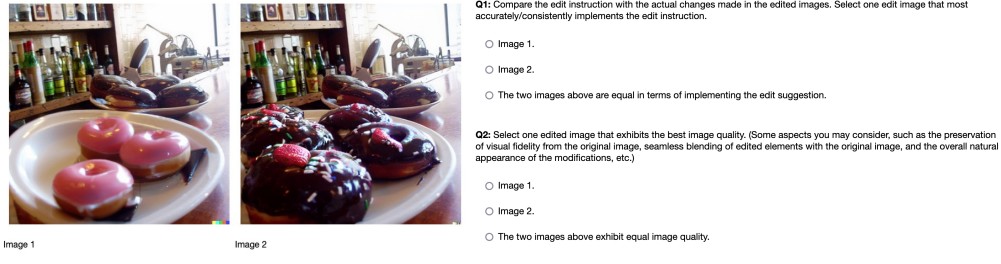

Figure 12: The interface of one-on-one comparison on AMT to assess generated images by different models.

