# OpenReview forum: "MagicBrush: A Manually Annotated Dataset for Instruction-Guided Image Editing"
_NeurIPS.cc/2023/Track/Datasets_and_Benchmarks — NeurIPS 2023 Datasets and Benchmarks Poster_

### Official Review · Reviewer_Sjyy · 2023-07-21
**DALL-E will not be surpassed**

**Rating:** 5
**Confidence:** 4
**Correctness:** Yes.
**Clarity:** Yes.

**Strengths:**

1. This is the first large-scale, manually annotated dataset for instruction-guided real image editing.
2. The dataset collection details are sufficient.
3. The paper is well-written and easy to follow.

**Additional Feedback:**

-

**Documentation:**

Yes.

**Ethics:**

No.

**Limitations:**

I did not find the place where the authors addressed the limitations.

**Opportunities For Improvement:**

1. Since the dataset is created by DALL-E, then the performance upper bound is DALL-E. So this dataset won't help improve the image generation quality for instruction-guided image editing.
2. Fine-tuning InstructPix2Pix on the dataset is not persuasive because it is sure to see a performance improvement.
3. It is not clear whether the test split of this dataset contains out-of-domain instructions which never appear in the training dataset. If there are, then what is the percentage?
4. The dataset quality is questionable. For example, in Fig. 5, after the "Dress him in a dinner jacket" instruction, the pose of the right arm changes obviously which is not instructed to do so.

**Relation To Prior Work:**

Yes.

**Summary And Contributions:**

The authors propose an instruction-guided image editing dataset with 10K manually annotated triples (source image, instruction, target image). The dataset supports multiple editing scenarios: single-turn, multi-turn, mask-provided, and mask-free editing. The authors fine-tune InstructPix2Pix on the new dataset, and achieves better performance.

---

> ### Author Response · Authors · 2023-08-21
> **Reply to Reviewer Sjyy**
>
> We appreciate your detailed and in-depth comments.
>
> > 1. Image Quality Upper Bound
>
> Thanks for bringing up this important point. Please refer to General Response 2. In short, the main challenge of instruction-guided image editing is not the “quality” of the final image *per se*, e.g., how photorealistic the image is. That is indeed bounded by DALL-E 2 (though we found the upper bound is actually quite high if one makes sufficient careful attempts). Instead, the main challenges are 1) whether the edits on the image are semantically consistent with the textual instruction, and 2) whether the edited parts blend seamlessly with their surroundings in the image. This is the main value proposition of our dataset: it provides high-quality supervision, through our carefully designed human-in-the-loop annotation process, for models to learn to overcome the aforementioned challenges. Such high-quality supervision for instruction-guided image editing simply didn’t exist before (e.g., InstructPix2Pix uses synthetic examples, which has a much lower signal-to-noise ratio).
>
> > 2. Expected Performance Improvement
>
> The reviewer is right that this improvement is expected, but that’s exactly the evidence we need to support our main claim and to show the value of this dataset. Our dataset’s main purpose is to improve the instruction-guided image editing capability (of both existing and future models) on real images, and that’s exactly what we showed through this experiment. Experiments with InstructPix2Pix is especially meaningful because the model has already been trained on a large amount of *synthetic* instruction-guided image editing data. Further improvement on that firmly confirms our dataset’s value in enhancing models’ editing capability on real images.
>
> > 3. OOD Instruction Percentage
>
> Thanks for bringing it up. Throughout the annotation process, we strongly encourage the workers to use diverse instructions by explicitly stating that in the tutorial and frequent email communications. We rejected some repetitive or trivial edits to ensure the diversity.
> As a result, the overlap between test instructions and training instructions is reasonably small. Since the definition of OOD is vague in this context, we show two numbers under different OOD definitions:
> If we consider exact match and deem a test instruction as OOD if it is not contained in training *verbatim*, then the OOD ration of the test set is 92.2%. However, this may be too strict so it serves as an upper bound.
> If we create a “signature” of each instruction by extracting its main verb and object class, e.g., (remove, cat), and use that as the basis for OOD definition instead, then the OOD ratio of the test set would be around 54%. However, this may be too loose because it may disregard important modifiers (e.g., “remove the cat *under the table*”), so it serves as a lower bound.
> In reality, the true OOD rate likely lies in between, depending on how one defines OOD.
>
> Finally, it’s worth noting that even when given the same edit instruction, the actual edits on different input images could be quite different. For example, for the same instruction *“remove the people in the background”*,  a model needs to generate different backgrounds based on the different input images.
>
> > 4. Dataset Quality
>
> Thanks for being so attentive. Admittedly, these unexpected changes are not desired and we have noted this in the revision (Section B.1 in the supplementary material).
> We would like to highlight the following points:
> - These examples only constitute a very small portion of the dataset, accounting for less than 2% (5 of the 300 edit turns that we manually checked). Therefore, they would not significantly impact the quality of our dataset.
> - These changes are usually minor and the main edit still follows the instruction, so it could still provide valuable supervision for a model to learn to better follow instructions.

---

> > ### Comment · Reviewer_Sjyy · 2023-08-25
> > **InstructPix2Pix needs to be trained from scratch on the proposed dataset**
> >
> > I think that in order to be persuasive that the dataset helps a method become better, InstructPix2Pix needs to be trained from scratch on the proposed dataset.
> >
> > I need to see evidence for the following claim:
> > "These examples only constitute a very small portion of the dataset, accounting for less than 2% (5 of the 300 edit turns that we manually checked). Therefore, they would not significantly impact the quality of our dataset."

---

> > > ### Author Response · Authors · 2023-08-29
> > > **Reply to Reviewer Sjyy**
> > >
> > > Thanks for the reply and suggestions.
> > >
> > > > 1. Training from Scratch
> > >
> > > Training from scratch is not the only way to help a method become better. Many datasets with relatively small size in both CV [1,2,3]  and NLP [4] domains mostly play their roles in the fine-tuning stage, yet they remain highly valuable to the community, especially when they also serve as evaluation benchmarks like our MagicBrush.
> > >
> > > The role of our dataset is also similar to the instruction tuning datasets for large language models (LLMs) [5, 6]. After massive pre-training, fine-tuning LLMs on these small instruction tuning datasets (usually a few dozens of thousands of examples) has proven critical for making them meaningfully following user instructions. The LIMA paper [7] showed that the quality of such instruction tuning data is of utmost importance, and using 1,000 *high-quality* examples was better than using much more mixed-quality data. In that regard, our manually annotated dataset, despite containing some noise, is much better in quality than the existing synthesized datasets. It provides unique values for making image editing models more meaningfully follow user instructions, for which we have shown through experiments with two image editing models, InstructPix2Pix and HIVE.
> > >
> > > Please refer to Reply to Reviewer 6Wso, fine-tuning with our dataset helps another method that just became available recently (HIVE) with a different model architecture, and the improvement based on human evaluation is even larger than that for InstructPix2Pix. This also supports our claim: our dataset helps instruction-guided editing methods become better at following instructions.
> > >
> > >
> > > [1] Learning by Planning: Language-Guided Global Image Editing, CVPR, 2021
> > >
> > > [2] Benchmark for Compositional Text-to-Image Synthesis, NeurIPS, 2021
> > >
> > > [3] The Dollar Street Dataset: Images Representing the Geographic and Socioeconomic Diversity of the World, NeurIPS, 2022
> > >
> > > [4] Natural Questions: a Benchmark for Question Answering Research, TACL, 2019
> > >
> > > [5] Stanford Alpaca: An Instruction-following LLaMA model, GitHub, 2023
> > >
> > > [6] Training language models to follow instructions with human feedback, arXiv, 2022
> > >
> > > [7] LIMA: Less Is More for Alignment, arXiv, 2023
> > >
> > >
> > > > 2. Evidence for Data Quality
> > >
> > > As shown in Tables 4, 5 and 6 of the paper and in reply to Reviewer 6Wso, our results demonstrate that fine-tuning with MagicBrush significantly improves the original methods (both InstructPix2Pix and HIVE), in terms of both edit instruction consistency and image quality.
> > >
> > > Additionally, we conducted an individual evaluation on HIVE and fine-tuned HIVE with respect to Edit Consistency and Image Quality. We randomly sample 100 edit turns in the test set for both HIVE and Fine-tuned HIVE, and invite 5 AMT workers to evaluate the instruction consistency and image quality. Each worker received an equal share of the generated images, specifically evaluating 40 images in total, with 20 images from HIVE and fine-tuned HIVE, respectively. Average scores are as follows:
> > >
> > > |               | Edit Instruction Consistency | Image Quality  |
> > > | ------------- | ------------- | ----- |
> > > | InstructPix2Pix | 3.0 | 3.2 |
> > > | Fine-tuned Pix2Pix | **3.1** | **3.6** |
> > > | | | |
> > > | HIVE | 2.4 | 2.7 |
> > > | Fine-tuned HIVE | **3.0** | **3.5** |
> > >
> > > The results align with the observation in the paper: our method can considerably improve both the image quality and edit instruction consistency.
> > >
> > > This again shows that, despite not being perfect and consisting of some noisy data (e.g., over-editing) in MagicBrush, fine-tuning on our dataset still improves the instruction following capability of existing image editing models, sometimes substantially.

---

### Official Review · Reviewer_xGdT · 2023-07-21
**MagicBrush Review**

**Rating:** 7
**Confidence:** 4
**Correctness:** The claims are correct.
**Clarity:** The paper is well-written and easily …

**Strengths:**

- Writing is clear, professional, and easy to understand
- High-quality dataset of great significance in the field
- Strong evaluation results and thorough experiments

**Additional Feedback:**

I believe to make the manuscript more easily understandable for trained readers not familiar with this particular sub field, it would be beneficial to explain how annotations are usually done with powerful neural networks instead of manually through professional software.

**Documentation:**

These aspects are appropriately discussed.

**Ethics:**

There are no ethical concerns from my perspective regarding MagicBrush, apart from the potential forgery misuse mentioned above.

**Limitations:**

This is a powerful technique that may enable advanced forms of forgery. Therefore, it is necessary that the authors adequately address the negative social impacts of this dataset in the main text, admittedly that this is discussed in section B in the appendix.

**Opportunities For Improvement:**

- It is slightly confusing why MagicBrush is described as "real image editing" since DALLE-2, a generative model, is involved. "Photo-realistic" might be a better phrasing here."
- To the point above, the role of DALLE-2 should be more emphasized in the diagrams (e.g. Figure 2), as text-driven editing using a generative model is confusing when mixed with the phrasing of "Manually Annotated" and "real image"

**Relation To Prior Work:**

Relationships to prior works are adequately and clearly discussed.

**Summary And Contributions:**

MagicBrush is a manually annotated dataset for instruction-guided photorealistic editing using DALLE-2 as the main editing platform. The dataset features 10k records of (source image x instruction x target image) triples and features benchmarks with fine-tuned InstructPix2Pix (state-of-the-art method in end-to-end image editing) and human evaluations to demonstrate the quality of this dataset over its predecessors.

---

> ### Author Response · Authors · 2023-08-21
> **Reply to Reviewer xGdT**
>
> We appreciate your positive comments and valuable feedback.
>
> > 1. About “Real Image Editing”
>
> We adopt this term from the literature [1, 2], where “real image editing” refers to that the editing *starts* with a real image. Technically, any subsequent image obtained by editing a real image, either by manual photoshopping or automatically by a model, should not be called real images anymore. We will make this distinction more clear.
>
> [1] Kawar et al., Imagic: Text-Based Real Image Editing with Diffusion Models, CVPR, 2023.
>
> [2] Zhu et al., In-domain gan inversion for real image editing, ECCV, 2020.
>
> > 2. Emphasis on the Role of DALL-E 2 (Improvement 2 and Additional Feedback)
>
> Thanks for the suggestion. In terms of using DALL-E 2, please refer to General Response 1 and we have updated Figure 2 and attached the detailed tutorial in the revision.

---

### Official Review · Reviewer_bheo · 2023-07-21
**Nice paper, but needs some details on data annotation and evaluation.**

**Rating:** 7
**Confidence:** 4
**Correctness:** I think there is no major problem wit…
**Clarity:** the paper is well wirtten.

**Strengths:**

- the paper introduces a relatively well-established data collection method based on image generation models, I think it benefits the application of learning-based image editing research.

- the results of several sota methods are analyzed to demonstrate the features of each model.

- data access is available.

**Additional Feedback:**

- the metrics in table 2 should cite references.

**Documentation:**

yes

**Ethics:**

ethical concerns are described in supplementary Section B.

**Limitations:**

See Opportunities For Improvement.
and the limitations are described in Supplementary Section B.

**Opportunities For Improvement:**

some details on data annotation and evaluation are required:

- the details of prompt instruction and overview keywords are provided. However, are the text descriptions predetermined for workers? or is it freely decided by the workers themselves? Are these text descriptions often used in image editing? is there any evaluation of it?

- 3.3 Data Quality Evaluation. Each worker evaluates 100 (500/5) edit turns? or all 500 edit turns? Is there any difference in the evaluation among different workers?

- is the data balance of different keywords considered for data splits.

- Section 4.3 Individual Evaluation. how many evaluators? how many images are evaluated? Are differences between raters taken into account?

**Relation To Prior Work:**

table 1 discussed this work differs from the previous datasets.

**Summary And Contributions:**

This paper introduces a text-guided image editing dataset based on DALL-E 2 platform, it consists of 5,359 sessions and 10,503 turns, supporting editing scenarios: single-/multi-turn, mask-provided, and mask-free. Their contributions are:

- a large number of text-guided image editing data, with good descriptions of dataset construction and prompt instruction.
- several SOTA methods are evaluated and analyzed on this dataset.

---

> ### Author Response · Authors · 2023-08-21
> **Reply to Reviewer bheo**
>
> Thanks for your valuable suggestions.
>
> > 1. Annotation Clarification
>
> Please refer to General Response 2. The description refers to the caption describing the entire image, which is required by DALL-E 2 for generation, while the instruction is concise and expressive.
>
> Since our focus is an instruction-guided editing dataset with high-quality images, the description can be freely decided by the workers as long as the target images are photorealistic and semantically consistent with the edit instructions.
>
>
> > 2. Data Evaluation Clarification
>
> 500 edit turns during the human evaluation phase are distributed equally among 5 workers. Each worker was responsible for evaluating 100 edit turns. The reported results are based on the average scores obtained from these evaluations.
>
> With the thorough check and clean mechanism mentioned in General Response 3, data quality remains at the same high standard. We invited five workers and calculated the average score to mitigate the impact of individual differences among workers.
>
> > 3. Data Balance Clarification
>
> That’s an interesting idea. We considered that before and decided to conduct random sampling eventually. This helps align the test set with the real-world edit distribution more. The main logic behind this decision is that edit instructions are naturally imbalanced (long-tailed) in reality. Therefore, balanced evaluation may draw some conclusions that are not applicable in real-world scenarios due to the keyword distributional gap.
>
> > 4. Evaluators Clarification
>
> During the individual evaluation, we invited five workers to participate and randomly selected 100 image examples as source images. As four models were involved, this resulted in a total of 400 generated images for evaluation. To ensure fairness and balanced evaluation by different workers, each worker received an equal share of the generated images, specifically evaluating 80 images in total, with 20 images from each of the four models.
>
> > 5. Metric Citation
>
> Thanks, we included citations of DINO and CLIP metrics in Section 4.1 Evaluation Metrics.

---

> > ### Comment · Reviewer_bheo · 2023-08-24
> > **Thanks for the reply**
> >
> > Thanks for the reply.
> > I changed my rating from 6 to 7.

---

> > > ### Author Response · Authors · 2023-08-24
> > > **Thanks for your feedback**
> > >
> > > Thank you once again for taking time to review our paper and for providing thoughtful suggestions.
> > > If you have any further questions or concerns, please feel free to comment, and we would be happy to discuss them.

---

### Official Review · Reviewer_6Wso · 2023-07-21
**A good paper with some minor issues**

**Rating:** 6
**Confidence:** 3
**Correctness:** Yes, the evaluation methods and exper…

**Strengths:**

1. The paper presents 10K manually annotated triplets (source images, instructions, target images), which can be utilized to train text-guided image editing models.
2. The proposed MAGICBRUSH dataset provides multi-turn capabilities compared with previous datasets.
3. The authors conduct experiments on some image-editing methods using the proposed dataset and provide quantitative results.

**Additional Feedback:**

Please refer to the previous questions.

**Clarity:**

The English writing should be further polished. For instance, in line 9 of the abstract, "annotated triples" should be corrected to "annotated triplets," and the same correction should be applied throughout the rest of the paper.

**Documentation:**

The authors provide a URL to access part of the dataset.

**Ethics:**

To my knowledge, there are no ethical issues with this paper.

**Limitations:**

Please refer to the previous question.

**Opportunities For Improvement:**

1. Please provide more details about the process of crowd worker annotation, including the total amount of time and budget required. Additionally, clarify what kinds of detailed tutorials were used, as mentioned in line 125.
2. It appears that the edited images all come from DALLE-2. The process of generating MAGICBRUSH seems more like data distillation of DALLE-2 results. In this case, could there be bias in the dataset, as they all come from the same method?
3. The results of DALLE-2 sometimes exhibit incorrect outcomes, as seen in the first row of Figure 1, where the people in the background of the county fair appear unnatural. The human face in the ground truth of MagicBrush in Figure 5 also looks unusual. Given these issues, using such images in MAGICBRUSH may introduce a significant gap between these images and real-world images in the wild. How were such issues addressed?
4. More methods, other than InstructPix2Pix2, should be evaluated using MagicBrush to further validate their effectiveness.
5. As mentioned by the authors, human evaluations were conducted to assess consistency and image quality. However, it seems that only 500 images in total were evaluated. Would this be sufficient to evaluate the entire dataset effectively?
6. Further discussion of the limitations and failure cases of MAGICBRUSH should be included.

**Relation To Prior Work:**

The paper provides sufficient comparison with prior works.

**Summary And Contributions:**

This paper introduces MAGICBRUSH, which comprises 10k manually annotated real image editing triplets designed to facilitate training large-scale text-guided image editing models. The authors further fine-tune instructpix2pix using this dataset and present evaluation results.

---

> ### Author Response · Authors · 2023-08-21
> **(1/2) Reply to Reviewer 6Wso**
>
> We appreciate your thoughtful and constructive comments.
>
> > 1. Time and Budget Details
>
> We reported the average annotation time and total cost in supplementary material B.4. Reiterating here with more details:
>
> Regarding time, the entire annotation process took around four months, including the time spent on iteratively refining the task design. The duration of an annotation session varies from 4 to 8 minutes, depending on the number of edit turns performed by the worker within the session. This results in a total annotation time of approximately 529 worker hours. As for the total budget/cost, we spent approximately \\$11,000, which includes the payments made to crowd workers on AMT (\\$8,000) as well as DALL-E 2 API (\\$3,000) costs. We have added these details to the appendix in the revision.
>
> > 2. Potential Bias from Distilling DALL-E 2
>
> Using DALL-E 2 is one of the key design choices we deliberated over when starting this work (details in General Response 2)
>
> There are two additional aspects that minimize potential biases in our data collection:
> - **The instructions are very diverse**. We strongly encouraged and carefully guided crowd workers (through frequent email communications) to increase instruction diversity. Different workers may also have varied prompt styles and preferences.
> - **The input images from MS COCO are diverse**. We further improve the diversity by balancing sampled images w.r.t. object classes (Sec 3.2). Since the edited images are required to be *natural*, i.e., the edited parts need to blend seamlessly with the unedited parts, the diversity of the original images is largely preserved in the edited images, further reducing potential biases.
>
> That being said, we acknowledge that inherent biases are hard to eliminate entirely. We will stay alert to potential biases in our dataset identified by the community and take prompt rectification actions. We have added more discussion about this in the revision (Section B.2  in supplementary material).
>
> > 3.  Unnatural Images
>
> Thanks for bringing this up. There are indeed some edited images that are not entirely natural, which is reflected in the image quality score from human evaluation. However, such images only constitute a small portion of the dataset (< 5%), and we believe it won’t lead to a significant gap. To further mitigate the impact of such imperfections in the data, a promising future direction is to explore ideas based on self-training and GAN: start with a model trained on our dataset, and iteratively apply the model to more input images and have a critic to judge and provide feedback on the generated images (e.g., based on how natural they are), which generates silver data to retrain the model. In this case, our dataset makes such a process more economical or even feasible by providing a much better initial model that significantly increases the signal-to-noise ratio of this process. We have added more discussion on this in the revision (Section B.1 in supplementary material).

---

> > ### Author Response · Authors · 2023-08-21
> > **(2/2) Reply to Reviewer 6Wso**
> >
> > > 4. More Methods Trained/Evaluated on MagicBrush
> >
> > Thanks for the suggestion. We totally agree that would strengthen the evaluation. We considered all existing instruction-guided image editing methods, including Inst-Inpaint [1] and InstructP2P [2], but they either only support limited edit types (e.g., removal only) or focus on different edit scenarios (3D). The most applicable method is HIVE [3], but its training code was unavailable online at our submission time.
> >
> > We recently fine-tuned HIVE (built upon Stable Diffusion 2.1) on MagicBrush and released their quantitative and human evaluation results here. Also, we provide some qualitative comparisons on our updated website.
> >
> > ### Quantitative Results of HIVE fine-tuned on MagicBrush
> > Following the same setting in the paper, we have following results on the MagicBrush test set:
> > |Setting| Methods | L1$\downarrow$ | L2$\downarrow$ | CLIP-I$\uparrow$ | DINO$\uparrow$ | CLIP-T$\uparrow$ |
> > | ------- | ------- | -------------- | -------------- | ---------------- | -------------- | ---------------- |
> > |Single-turn| HIVE | 0.1092 | 0.0341 | 0.8519 | 0.7500 | 0.2752 |
> > || &nbsp;&nbsp;w/ MagicBrush | **0.0658** | **0.0224** | **0.9189** | **0.8655** | **0.2812** |
> > |Multi-turn| HIVE | 0.1521 | 0.0557 | 0.8004 | 0.6463 | 0.2673 |
> > || &nbsp;&nbsp;w/ MagicBrush | **0.0966** | **0.0365** | **0.8785** | **0.7891** | **0.2796** |
> >
> > ### Human Evaluation Results of HIVE fine-tuned on MagicBrush
> > We conducted one-on-one comparisons for HIVE vs. HIVE-tuned and HIVE-tuned vs. GT (Ground truth from MagicBrush).
> > Specifically, we randomly sampled 100 edit turns from the test set for each comparison and divided the data equally among 5 Amazon MTurk workers for evaluation.They were asked to select the better option in terms of the Instruction Consistency and Image Quality:
> >
> > |               | Consistency |       |
> > | ------------- | ----------- | ----- |
> > | Fine-tuned HIVE | HIVE | Tie |
> > | **51** | 16             | 33   |
> > | Fine-tuned HIVE | GT | Tie |
> > | 12 | **69** | 19 |
> >
> > |               | Image Quality |       |
> > | ------------- | ------------- | ----- |
> > | Fine-tuned HIVE | GT | Tie |
> > | **58** | 19            | 23   |
> > | Fine-tuned HIVE | GT | Tie |
> > | 20 | **61** | 19 |
> >
> >
> > From these two tables, we have observations that are consistent with the paper:
> > - Fine-tuning HIVE on MagicBrush boosts their performances by a large margin.
> > - Compared to the MagicBrush ground truths, current methods still have substantial room for improvement.
> >
> > We will update the revision with these results in the final camera-ready version and move the current rebuttal page (Page 10) to appendix.
> > We'd like to train/evaluate other models whenever available. Also, we’d welcome any suggestions of other available methods that we may have missed.
> >
> >
> > [1] Yildirim et al., Inst-Inpaint: Instructing to Remove Objects with Diffusion Models, arXiv, 2023.
> >
> > [2] Xu et al., InstructP2P: Learning to Edit 3D Point Clouds with Text Instructions, arXiv, 2023.
> >
> > [3] Zhang et al., HIVE: Harnessing Human Feedback for Instructional Visual Editing, arXiv, 2023.
> >
> > > 5. Scale of Dataset Quality Evaluation
> >
> > Thanks for the question, we believe 500 random examples are sufficient for data quality evaluation.
> > As detailed in General Response 3, throughout the annotation process, we perform regular checks on data quality and take actions if necessary. Therefore, we believe the data quality remains at the same high level and evaluating on a subset with 500 examples would be sufficient.
> >
> > > 6. Limitations and Failure Cases
> >
> > Thanks, we mentioned some of the limitations in B.1 and we have added more limitations such as a small portion of unnatural images in the updated revision.
> >
> > > 7.  Triple vs. Triplet
> >
> > Thanks for the suggestion. We used “triple” because it is widely used in NLP, especially in the knowledge graph literature where the term is frequently used. We have changed `triple` to `triplet` in the revised version and will further carefully polish the writing.

---

### Author Response · Authors · 2023-08-21
**General Response**

We’d like to thank all reviewers for taking the time and effort to rigorously evaluate our work. We'll take the suggestions for improvements seriously and try to address each concern.

## 1. Tutorial Details of Using DALL-E 2
(Reviewer 6Wso #1 and xGdT #2)

Thanks for the suggestion, we included the screenshot to illustrate how to edit step by step in the previous supplementary material.

To provide more details, now in both the updated submission (Page 10) and supplementary material (Section E.1), we include how to log into DALL-E 2 and conduct mask drawing, prompt engineering, and data selection to get satisfactory target images.
Also, we include the link to the video used as part of the tutorial in both places.


## 2. Distillation of DALL-E 2 and Performance Upper Bound
(Reviewer 6Wso #2 and Sjyy #1)

Thanks to reviewers for bringing up this important point. Using DALL-E 2 is one of the key design choices we deliberated over when starting this work.
Through preliminary exploration on various text-to-image models including Stable Diffusion, we found DALL-E 2 is pretty powerful, if one is willing to spend sufficient time and money on it. That is, given an instruction (e.g., *“remove the wooden frame”* in Figure 1) and an input image, if one makes sufficiently many attempts with DALL-E 2, including drawing proper masks, coming up with different descriptions (e.g., *“this is a cabinet for toy display with no frames”*), and letting DALL-E 2 generate multiple times, it’s likely that one will find a generation that satisfies both the semantic consistency and image quality requirements.
In fact, DALL-E 2, can not edit images itself, due to the requirement of user-provided masks and global descriptions.

Therefore,  DALL-E 2, has a fairly high upper bound in its image generation capability—it can generate diverse edits that are consistent with the instruction, under proper and sufficient user guidance. **This is exactly what our human-in-the-loop data collection does**.


## 3. Data Quality Check during Annotation

**During the entire annotation process, we continuously monitored it to ensure data quality.**
Specifically, in the trial period, we checked all annotated examples in a batch with 10 sessions to provide prompt feedback to each worker on data quality (both in image and instruction). Only workers who can deliver satisfactory results will be advanced to the next stage, where they will be asked to do more tasks on AMT. Then, for every 100 sessions, we spot-checked 5 of them throughout the annotation process. Sessions containing subpar images, with issues related to image quality or instruction consistency, are removed.

Additionally, we maintained frequent communication with the workers and provided timely guidance, and requested certain turns to be redone if the quality is unsatisfactory. As time progresses, we observe a significant decrease in the frequency of communication that was needed, and we find that all workers consistently pass the checks in the later batches of data annotations. This indicates a notable improvement in the quality of the annotated data as the process advances. Meanwhile, the results of our human evaluation task on 500 random sampled edit turns reveal average scores of 4.1 and 3.9 out of 5.0 for consistency and image quality, respectively. Compared to edited images by existing methods in Section 4.4, these scores demonstrate the high quality of our dataset.


We have added such annotation details in the revision (Section E.1 in the supplementary material).

---

### Decision · Program_Chairs · 2023-09-22

**Decision:**

Accept (Poster)

**Comment:**

This paper introduces a text-guided image editing dataset based on DALL-E 2 platform, it consists of 5,359 sessions and 10,503 turns, supporting editing scenarios: single-/multi-turn, mask-provided, and mask-free.

The reviewers mostly acknowledge the contribution of the dataset. They also raised concerns on data bias caused by DALL-E2, performance upper bound, and data quality control. The AC found the rebuttal mostly solved the reviewers' concerns. The authors are encouraged to further improve their dataset and show details of the annotation process to convince wide adoptation of the dataset.